# Sparse Autoencoders Reveal Temporal Difference Learning in Large Language Models

**Can Demircan**[*,1]
can.demircan@helmholtz-munich.de

**Tankred Saanum**[*,2]
tankred.saanum@tuebingen.mpg.de

**Akshay K. Jagadish**[1,2]     **Marcel Binz**[1]     **Eric Schulz**[1]

## Abstract

In-context learning, the ability to adapt based on a few examples in the input prompt, is a ubiquitous feature of large language models (LLMs). However, as LLMs' in-context learning abilities continue to improve, understanding this phenomenon mechanistically becomes increasingly important. In particular, it is not well-understood how LLMs learn to solve specific classes of problems, such as reinforcement learning (RL) problems, in-context. Through three different tasks, we first show that Llama 3 70B can solve simple RL problems in-context. We then analyze the residual stream of Llama using Sparse Autoencoders (SAEs) and find representations that closely match temporal difference (TD) errors. Notably, these representations emerge despite the model only being trained to predict the next token. We verify that these representations are indeed causally involved in the computation of TD errors and $Q$-values by performing carefully designed interventions on them. Taken together, our work establishes a methodology for studying and manipulating in-context learning with SAEs, paving the way for a more mechanistic understanding.

## 1 Introduction

Large language models (LLMs) pretrained on large-scale text corpora are proficient in-context learners. They can predict and learn novel rules and functions conditioned only on a text prompt with a few examples. This phenomenon has been demonstrated under many different paradigms, such as translation (Brown et al., 2020; Wei et al., 2022), function learning and sequence prediction (Garg et al., 2022; Coda-Forno et al., 2023), and even reinforcement learning (Binz & Schulz, 2023; Shinn et al., 2024; Brooks et al., 2022; Schubert et al., 2024; Hayes et al., 2024). Consequently, understanding *how* in-context learning is implemented mechanistically becomes increasingly important. In this paper, we investigate the mechanisms underlying in-context reinforcement learning (RL) in LLMs, specifically how they can learn to generate actions that maximize future discounted rewards through trial and error, given only a scalar reward signal.

In RL, the central learning signal is the temporal difference (TD) error (Sutton, 1988). The TD error is the difference between the agent's belief about a state-action pair's value, and a target value, which is constructed as the immediate reward plus the discounted value of the successor state. For instance, when the agent receives an unexpected reward, or unexpectedly transitions to a high-value successor state, the corresponding TD error can be used to update beliefs about the value of the current state. This learning rule forms the foundation of many RL algorithms, such as $Q$-learning (Mnih et al., 2015; Watkins & Dayan, 1992) and Actor-Critic (Konda & Tsitsiklis, 1999; Sutton et al., 1999; Haarnoja et al., 2018). TD learning has also played an important role in neuroscience, where several experiments have shown links between TD-like computations and dopamine in multiple species

---

∗ Equal contribution.

[1]Institute for Human-Centered AI, Helmholtz Computational Health Center, Munich, Germany

[2]Max Planck Institute for Biological Cybernetics, Tübingen, Germany

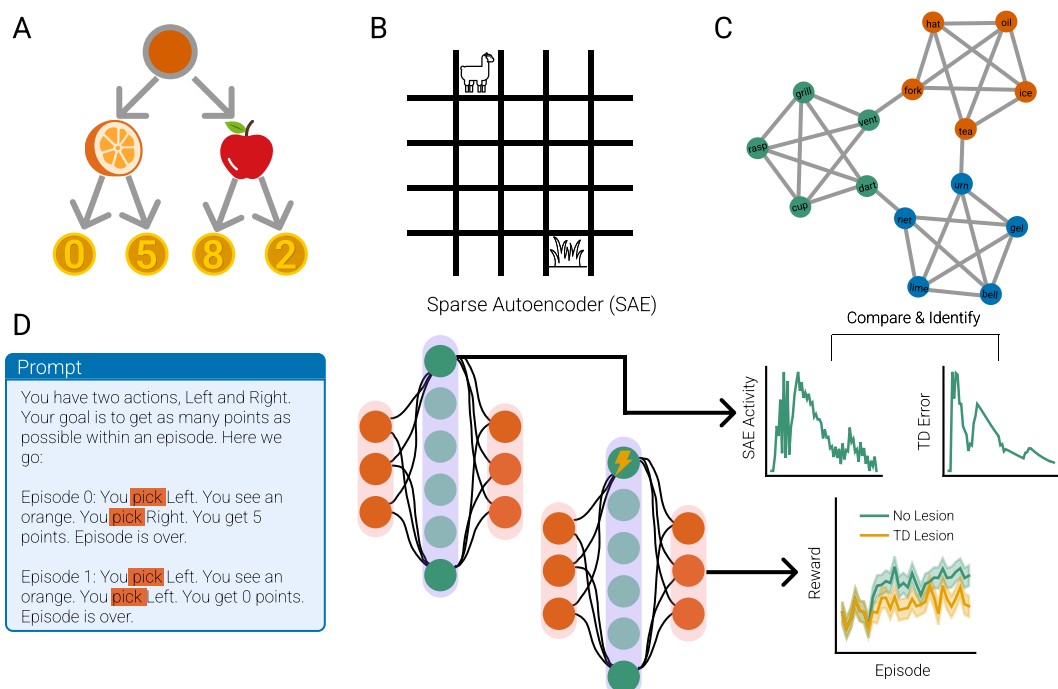

Figure 1: We study the mechanisms of in-context learning in three different tasks: (*A*) The Two-Step Task, (*B*) the Grid World task, and (*C*) the graph prediction task. (*D*) Example pipeline for the Two-Step Task. We prompt Llama as shown on the left. As it selects its actions, we record the internal representation for the tokens that precede the actions, which are highlighted in orange. We train SAEs on these representations (middle) and correlate with the learned latents of the SAEs against the TD error signals and other variables of interest we obtain from reinforcement learning agents (right). After identifying such latents, we lesion[2] them and replace Llama's internal representations with reconstructions following the lesions (middle). We then test whether the lesion created the expected effects in the behavior (right).

(Montague et al., 1996; Schultz et al., 1997; Fiorillo et al., 2003; Flagel et al., 2011; Niv et al., 2015).

TD learning is a fundamental algorithm that has been successful in modeling both machine and animal learning. Could LLMs learn to implement TD learning in-context simply from being trained to perform next-token prediction? We use sparse autoencoders (SAEs) to analyze the representations supporting in-context learning in RL settings. These models have successfully been used to build a mechanistic understanding of neural networks and their representations (Bricken et al., 2023; Templeton et al., 2024; Gao et al., 2024; Lieberum et al., 2024). Across several experiments, we establish a methodology for systematically studying *and* manipulating in-context RL in Llama 3 70B[1] (Dubey et al., 2024) with SAEs (see Figure 1). Using our methodology, we uncover representations similar to TD errors and *Q*-values in multiple tasks. Moreover, we find that manipulating these features changes Llama's behavior and representations in predictable ways. We believe our study paves the way for a mechanistic understanding of in-context learning, setting a precedent for investigating how LLMs solve other types of learning problems in context.

---

[1]From here onwards, we refer to this model as Llama.

[2]Lesioning refers to setting the activations of specific units to 0. This is also commonly referred to as zero ablation in the literature (Heimersheim & Nanda, 2024).

## 2 METHODS

### 2.1 REINFORCEMENT LEARNING

We investigate in-context learning in the setting of Markov Decision Processes (MDPs). An MDP consists of a state space $\mathcal{S}$, an action space $\mathcal{A}$, and transition dynamics $s_{t+1} \sim T(s_{t+1}|s_t, a_t)$, defining the probability distribution of successor states given the current state and action, as well as a reward function $r(s_t, a_t)$ that maps state-action pairs to a scalar reward term. The goal of the agent is to learn a policy $\pi_\theta(a_t|s_t)$ that maximizes future discounted rewards, e.g. $Q$-values $Q_{\pi_\theta}(s_t, a_t) = \mathbb{E}_{\pi_\theta}\left[\sum_{t=0}^{T} \gamma^t r(s_t, a_t)\right]$, where $\gamma$ is a discount factor.

A canonical algorithm for learning the $Q$-value function for a fixed policy $\pi_\theta$, is TD learning. From an initial estimate of the $Q$-values, TD learning bootstraps a sequence of estimates that converge to the true values. This is done by minimizing *TD errors*, the difference between the agent's previous estimated $Q$-values, and the immediate reward plus the discounted $Q$-value of the successor state. Assuming the policy always picks the action that maximizes $Q$-values, the temporal difference error and subsequent $Q$-value update can be written as follows:

$$\delta_{\text{TD}} = r(s_t, a_t) + \gamma \max_a Q(s_{t+1}, a) - Q(s_t, a_t) \tag{1}$$

$$Q(s_t, a_t) \leftarrow Q(s_t, a_t) + \alpha\, \delta_{\text{TD}} \tag{2}$$

For subsequent analyses, we rely on $Q$-learning, which is a variant of TD learning that learns a value function for the optimal policy. See Appendix A.1 for details surrounding the implementation of the $Q$-learning model.

### 2.2 SPARSE AUTOENCODERS

A natural place to look for the traces of TD errors and $Q$-values is in the residual streams of the Llama's transformer blocks (Templeton et al., 2024; Bricken et al., 2023; Gurnee et al., 2023). The residual stream of a transformer block carries information from the input and all preceding blocks, making it a viable candidate for initial exploration.

Directly searching for features that correspond to $Q$-values or TD errors in the residual stream, while possible, is often impractical. In Llama, each token is represented as an 8192-dimensional vector in the residual stream. Due to the high number of statistical comparisons such an approach would necessitate, analyses showing connections between activations and TD errors will suffer from reduced power. Furthermore, it has been shown that LLMs represent concepts in an entangled and distributed manner. This phenomenon, known as polysemanticity (Elhage et al., 2022), occurs when single semantic concepts are represented across multiple neurons. Using tools from the field of mechanistic interpretability (Olah, 2023), we seek to learn a *disentangled*, *monosemantic*, and potentially lower dimensional set of features from the residual stream. We hypothesize that features corresponding to TD errors or action values can be found in this low-dimensional disentangled latent space.

One popular model for learning such features is the Sparse Autoencoder (SAE) architecture. SAEs are trained with gradient descent to reconstruct the potentially entangled features learned by models like transformers through a linear combination of sparse latent features. SAEs have been shown to learn more easily interpretable and monosemantic features (Cunningham et al., 2023; Bricken et al., 2023). We used SAEs with a single dense encoder layer ($f$), followed by a ReLU non-linearity ($\sigma$) and a single dense decoder layer ($g$). We train SAEs to reconstruct vectors from the residual stream $h$ while minimizing the $L_1$ norm of the encoded features:

$$\mathcal{L}_{\text{SAE}} = ||h - \tilde{h}||_2^2 + \beta||\sigma(f(h))||_1^2 \tag{3}$$

where $\tilde{h} = g(\sigma(f(h)))$. We trained separate SAEs for each of the 80 blocks of Llama and each experiment. Further training details are provided under Appendix A.2.

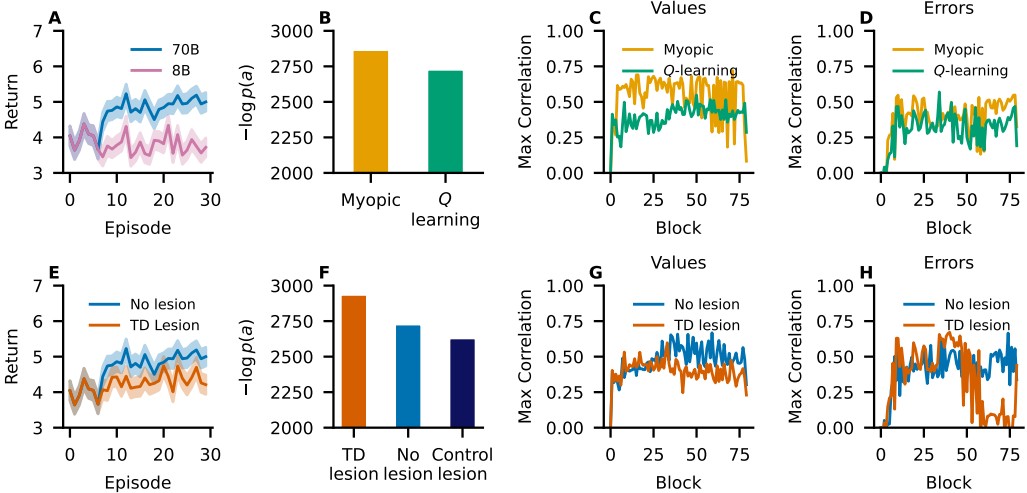

Figure 2: Llama relies on TD-like features to solve RL tasks in-context. (*A*) Llama 70B often learns the optimal policy in the Two-Step Task through trial and error, whereas the smaller 8B counterpart does not improve beyond chance level. Shaded regions show standard error of the mean. (*B*) Llama's behavior is best described by a $Q$-learning algorithm. (*C & D*) SAE features with significant correlations to both reward estimates (myopic values) and $Q$-value estimates, as well as temporal difference errors, appear gradually through the transformer blocks. (*E* and *F*) Deactivating a single TD feature in Llama is sufficient to impair performance and make behavior less consistent with $Q$-learning. (*G & H*) Negatively scaling the TD feature decreases subsequent representations' similarity to $Q$-values and TD errors.

## 3 LLAMA USES TD FEATURES TO LEARN POLICIES IN-CONTEXT

We first sought to assess Llama's ability to solve an RL problem purely in-context. We designed a simple MDP inspired by the Two-Step Task (Daw et al., 2011; Kool et al., 2016). In this task, Llama first had to choose to go `Left` or `Right` to enter either the `Apple` state or the `Orange` state, respectively. This first state-transition was always awarded with 0 rewards. From there, Llama again had to choose `Left` or `Right` to enter the terminal state and receive a reward. The reward and task structure are visualized in Figure 1A. Rewards and transition dynamics were deterministic. We instructed Llama to maximize reward and encouraged it to explore more at the beginning of the experiment (see Appendix A.3 for the exact prompt). When entering a new state, we sampled an action from Llama's predictive distribution of the possible action tokens with a temperature of 1. Llama completed 100 independent experiments initialized with unique seeds, each consisting of 30 episodes. We sampled actions from a random policy in the first 7 episodes to ease the exploration problem.

We first observe that Llama achieved significantly higher returns than chance in the Two-Step Task, and often learned the optimal policy (see Figure 2A). The smaller 8B parameter version of Llama performed roughly at chance level, suggesting that the ability to learn policies through RL emerges with scale. As a result, we focus on the 70B parameter version for subsequent analyses. Next, we fitted behavioral models to the sequences of actions performed by Llama. We fitted a $Q$-learning model that updated $Q$-values using TD errors, a myopic $Q$-leaning model that only cared about immediate rewards[2], which we implemented by setting the discount parameter $\gamma$ of the $Q$-learning model to 0, as well as a repetition model that computed choice probabilities as a function of how often an action was picked in a given state in the past. We see that Llama's behavior is best captured by a $Q$-learning model, attaining a negative log-likelihood (NLL) score of 2729, outperforming the myopic model with an NLL of 2864. This indicates that it integrates information about future

---

[2]In the animal learning literature, this is referred to as the Rescorla-Wagner model (Rescorla & Wagner, 1972).

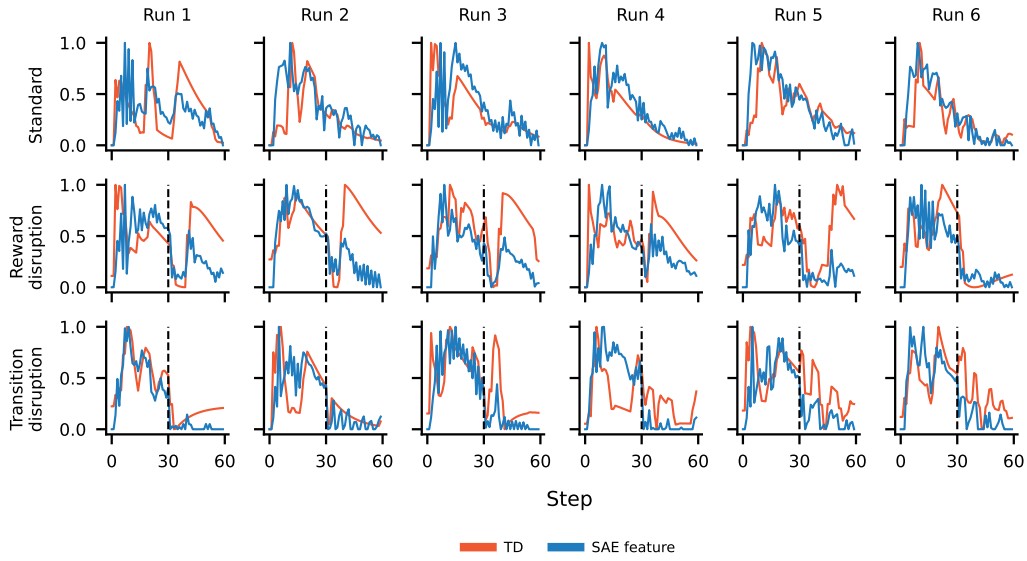

Figure 3: Qualitative comparison of the TD error and the best matching SAE feature from block 34 for three separate runs in all three variations of the Two-Step Task. The SAE feature shows similar jumps as TD when Llama encounters surprising events, such as transitioning to unexpected states or receiving an unexpected reward. The dashed line indicates the onset of the change in either transition dynamics or reward function.

discounted rewards to select actions (see Figure 2B). The repetition model, which predicted that Llama simply repeated action patterns in the prompt, fits behavior worse than chance with an NLL of 5745.

Next, we trained SAEs on the residual stream outputs from all of Llama's transformer blocks. To create a diverse training set for the SAEs, we collected representations from Llama on two additional variations of the Two-Step Task where either the reward function or the transition function changed in the middle of the experiment. Since unexpected changes to either transition dynamics or reward function will be followed by large positive or negative TD errors, we expected it to be useful for detecting possible TD errors in Llama's activations. We trained each SAE using a regularization strength $\beta = 1e-05$ for 30 epochs on 18000 residual stream representations. In the resulting SAEs, we find features gradually developing over the transformer blocks that track both the reward function (captured in the myopic values) as well as $Q$-values (see Figure 2C).

Furthermore, we observe features that show significant correlation with TD errors (see Figure 2D). In block 34, we find the SAE feature with the highest correlation ($r = 0.58$) with the TD error of our $Q$-learning model. We visualize the normalized SAE feature – overlaid with the normalized TD error – in Figure 3 for six different runs and the three variations of the Two-Step Task. The feature shows the same qualitative change as the TD error when the reward function and transition dynamics change. We refer to these types of latent features that resemble TD errors as *TD latents*. To establish whether this TD latent has a causal connection to RL behavior, we tested Llama on the Two-Step Task with the TD latent either deactivated or negatively scaled. Before each choice, we replaced the residual stream activations in block 34 with reconstructions from the trained SAE in which the TD latent was set to 0 or scaled. We see that simply deactivating the TD latent led to significantly worse performance in the task (see Figure 2E), producing behavior that differed more from a $Q$-learning agent (Figure 2F). In contrast, deactivating the SAE feature in block 34 with the *lowest* correlation to TD did not produce behavior that was less likely according to the $Q$-learning model. Finally, we analyzed correlations between SAE features and $Q$-values and TD errors *after* we scale the TD latent by a value of $-10$, similar to Templeton et al. (2024). Here we see a substantial decrease in the correlation strengths for both $Q$-values and TD errors (see Figure 2G & H). In sum, we found an SAE feature showing the characteristics of TD error across three different variations of the Two-Step

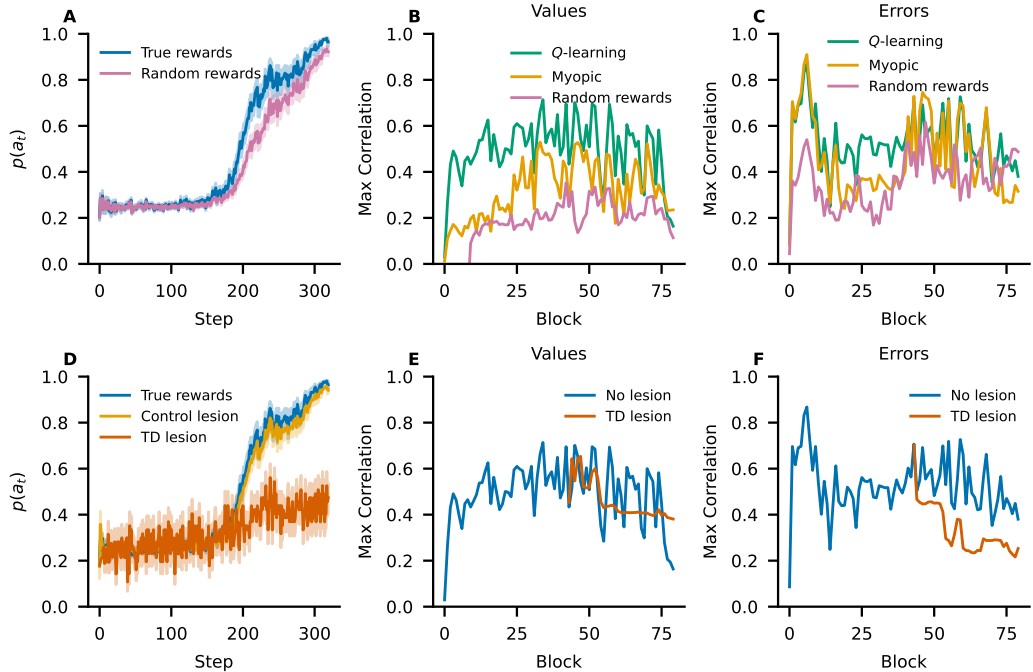

Figure 4: Llama can predict the actions of a $Q$-learning agent and keeps track of variables similar to $Q$-values and TD errors. (*A*) Llama predicts action sequences better when given correct information about rewards. (*B & C*) $Q$-values and a reward-tracking variable, as well as accompanying error signals, are significantly correlated with SAE features. Max correlations shown after Gaussian smoothing with $\sigma = 0.5$. (*D*) Lesioning the TD latents impacts action prediction accuracy, whereas lesioning other features barely affects action predictions. (*E & F*) Lesioning TD latents also impacts subsequent $Q$-value and TD error representations.

Task. Intervening on this single SAE feature was sufficient to significantly alter Llama's ability to perform RL in-context and impacted subsequent representations.

## 4 $Q$-VALUES AND TD SUPPORT OPTIMAL ACTION PREDICTION

In the previous task, we showed that Llama's behavior and internal representations in a simple RL task could be predicted by a $Q$-learning model. Next, we investigated whether we could also identify TD errors and $Q$-values in Llama when prompted by a sequence of observations and actions generated by a $Q$-learning agent. This setup allowed us to analyze Llama's representations in a more complex MDP. We trained $50$ separate $Q$-learning agents to navigate from a fixed starting location to a fixed goal location in a $5 \times 5$ grid using $4$ different actions (UP, DOWN, LEFT, and RIGHT) (Figure 1B). Entering the goal state led to $+1$ reward and terminated the episode, while entering any other state yielded $-1$ reward. Following Kaplan et al. (2020), we quantified Llama's in-context reinforcement learning ability by how well it could predict the $Q$-learning agent's actions over time. Further details about the task, including the prompt structure, are provided in Appendix A.4.

First, we observed that Llama improved at predicting the actions of the $Q$-learning agent as it received more observations. While this indicates in-context learning, we cannot yet attribute this ability to reinforcement learning, as Llama could simply learn the action sequence without attending to the reward. To control for this, we provided Llama with a second prompt, where the action sequences were identical, but the rewards were randomly set to $+1$ or $-1$, independently of whether the agent reached the goal. As shown in Figure 4A, Llama is better at predicting the same action sequences when given the correct rewards, indicating that it integrated reward information to predict actions. See Appendix A.4 for additional control analyses.

We then trained SAEs on Llama's residual stream representations for this task and examined whether the learned latents corresponded to the $Q$-values and the TD errors of the agent that generated the observations. The training was done for 15 epochs with a regularization strength of $\beta = 1e - 05$. As in the previous task, we correlated each latent representation learned by the SAEs with the $Q$-values, as well as the myopic values. As an additional control, we included estimates from a $Q$-learning agent trained on the random reward sequence from the control analysis presented above. Throughout Llama's transformer blocks, we observed the highest correlations against the $Q$-values that were used to generate the observations (Figure 4B). Similarly, most blocks showed the highest correlations for the TD errors of the same model. We also found high correlations with the myopic error signal, indicating that Llama learned rewards as well (Figure 4C). These findings further support the hypothesis that Llama's internal representations encode reinforcement learning-like computations, even in more complex environments with larger state and action spaces.

Lastly, we validated the causal role of the identified TD latents by deactivating and scaling them. We selected 4 different latents across 4 different blocks and set their activity to 0. All of these latents were selected from block 40 onwards and had $r \geqslant 0.75$ with the TD error. We selected more features to deactivate in this experiment to elicit stronger effects on Llama's behavior. We also conducted control lesions, where we set the activity of latent units from the same blocks that has the lowest correlation with TD to 0. We find that lesioning the TD latents significantly degraded Llama's ability to predict actions, whereas the control lesions led to very small changes in Llama's action predictions (Figure 4D). Deactivating or scaling the TD latents, we observed a decrease in both the $Q$-value correlations (Figure 4E) and the TD error correlations (Figure 4F). These results provide evidence for the causal role of the identified TD latents in Llama's reinforcement learning capabilities.

## 5 LEARNING GRAPH STRUCTURES WITHOUT REWARDS

So far, we have studied TD learning in the context of RL. However, TD learning can also be used to learn about statistical structures in the absence of rewards. For example, the Successor Representation (SR; Dayan, 1993) encodes an MDP in terms of future discounted state occupancies given the agent's policy, and can be learned using TD learning. The SR is represented as a state by state matrix $\mathbf{M}$, where $\mathbf{M}(s, s')$ is the future discounted expected occupancy of state $s'$ when the agent is at state $s$, e.g. $\mathbf{M}(s, s') = \mathbb{E}\left[\sum_{t=0}^{\infty} \gamma^t \mathbb{I}(s_t = s')|s_0 = s\right]$ where $\mathbb{I}$ is the indicator function. The TD error computation and the update defined in equation 1 & equation 2 can be generalized to learn this representation as follows:

$$\delta_{\text{TD}} = \mathbf{1}_{s_t} + \gamma \mathbf{M}(s_{t+1}, :) - \mathbf{M}(s_t, :) \tag{4}$$
$$\mathbf{M}(s_t, :) \leftarrow \mathbf{M}(s_t, :) + \alpha \, \delta_{\text{TD}} \tag{5}$$

Here $\mathbf{1}_{s_t}$ is a one-hot vector at state index $s_t$. The SR has seen wide application not only in RL research (Kulkarni et al., 2016; Barreto et al., 2017; Machado et al., 2017), but has also been used as a model of hippocampal function in neuroscience (Garvert et al., 2023; Gershman, 2018; Stachenfeld et al., 2017; Gardner et al., 2018). For example, it has been shown that when humans are presented with a sequence of stimuli that are generated from a latent graph, the representations in the medial temporal lobe resemble the diffusion properties of the graph (Schapiro et al., 2013; 2016; Garvert et al., 2017), which is captured by the SR.

If Llama uses TD-like computations to represent $Q$-values, does it similarly rely on a temporal difference error signal to learn something akin to the SR? To test this hypothesis, we prompted Llama with a sequence of observations generated from a random walk on a latent community graph (Schapiro et al., 2013) (Figure 1C). This graph had three communities, each community with five fully connected nodes. Each community also had two *bottleneck* nodes that connected to other communities. The model was prompted to predict the next observation at each time point. There were 401 observations, and we repeated this procedure 20 independent times, using randomly sampled node names.

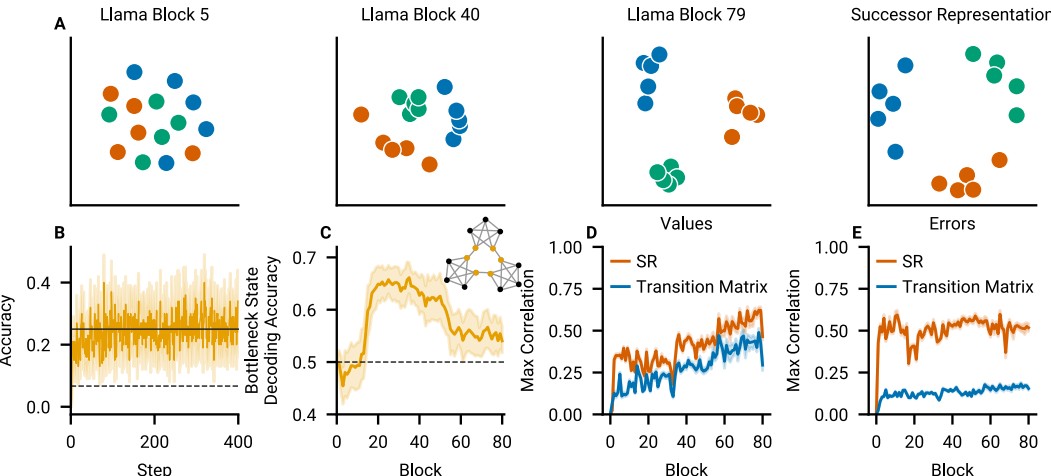

Figure 5: Llama learns graph structures through TD-learning, representing them similarly to the successor representation (SR). (*A*) Llama's state representations projected in 2D space, using multidimensional scaling, shows the emergence of latent graph structure across transformer blocks. (*B*) Llama quickly achieves high accuracy in predicting the next state. Accuracy is averaged over 100 runs. (*C*) Bottleneck states can be linearly decoded from middle blocks onward. (*D & E*) Latent representations of SAEs trained on Llama's representations strongly correlate with the SR and associated TD learning signals, outperforming model-based alternatives. Shaded regions in *B-E* indicate 95% confidence intervals. Dashed horizontal lines indicate chance level performance, and the solid horizontal line represents the ceiling.

First, we saw that Llama learns the local structure of the graph and can predict the next state with ceiling accuracy after around 100 observed state transitions (Figure 5B). Does Llama also learn the global connectivity structure of the graph? To test this, we performed multidimensional scaling (Torgerson, 1952) on all 80 residual streams' representation of each node in the graph after learning. We saw that the 2D projections of these representations gradually grow more and more similar to the 2D projection of the SR of the graph (Figure 5A), indicating that Llama represents the global geometry of the latent graph.

To further probe Llama's understanding of the global graph structure, we tested whether it explicitly represents bottleneck states. These are special states that allow transitions between communities and are particularly useful for planning in hierarchically structured environments. Although such representations are not needed to predict the next tokens, we found that they can be linearly decoded starting around block 20 onward (Figure 5C), with the decoding accuracy peaking around halfway through the model.

These findings show that Llama builds a representation of the graph's global geometric properties over blocks. However, this could be achieved without TD learning, such as through a model that learns the environment's transition matrix by counting experienced transitions. To test whether Llama acquires structural knowledge through TD learning, we trained SAEs on its representations and compared them to agents learning either the SR using TD errors, or simply the graph's transition matrix, respectively (see Appendix A.5 for training details).

We identified latents that were maximally correlated with each set of representations. We found stronger correlations with the SR (max $r = 0.62$) than with the transition matrix (max $r = 0.49$) throughout the model, as shown in Figure 5D. These representations build gradually and peak late in the model. Consequently, we found that the TD errors (max $r = 0.60$) that are used to learn the SR had much stronger correlations than a surprise signal corresponding to the log-likelihood of the next state under the learned transition matrix (max $r = 0.18$; Figure 5E). We additionally replicated these findings using more traditional methods such as representational similarity analysis (Kriegeskorte et al., 2008; Kornblith et al., 2019) in Appendix A.5.

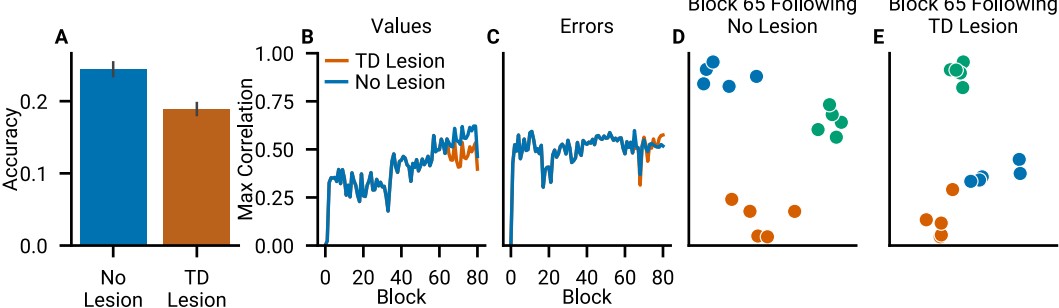

Figure 6: Lesioning TD SAE latents impair behavior and representations. (*A*) Following a lesion in block 64, Llama's accuracy in predicting the next state drops. (*B*) The SAE representations following the lesioning have reduced correlations with the SR, despite the recovery of the TD error following lesioning (*C*). While the community structure is reflected in the original representations in Block 65 (*D*), these representations are disrupted as a result of lesioning in earlier block (*E*).

Lastly, we deactivated the TD latent that had the strongest correlation with the TD error, which was in block 64. Following this intervention, Llama's prediction accuracy for the next state decreased from 24.3% to 18.9% (Figure 6A). This also reduced the subsequent correlations between the SAE latents and the SR (Figure 6B), despite a rapid recovery of the TD correlations after the lesion (Figure 6 C). The disruption of the learned structure can be seen in block 65, where the community structure of the graph is distorted (Figures 6D and 6E). Taken together, our behavioral and representational analyses show that Llama's graph knowledge is represented similarly to the SR, and the TD latents we identified support learning these representations.

# 6  RELATED WORK

Understanding in-context learning in transformers has received significant attention (Garg et al., 2022; von Oswald et al., 2023; Dai et al., 2023; Ahn et al., 2023). Previous work has shown that transformers can discover known and existing algorithms without explicit guidance towards these algorithms. Akyürek et al. (2023) have trained transformers on linear regression problems and shown that the model can implement ridge regression, gradient descent, or ordinary least squares in-context. Focusing on RL, Wang et al. (2024) have shown that linear transformers trained explicitly to solve reinforcement learning problems discover TD learning methods. On a larger scale, Laskin et al. (2022) has shown that in-context RL can be distilled to smaller models. We build on this by showing that in-context RL can emerge in LLMs simply as a consequence of language pre-training, even if the model was never explicitly trained to solve RL problems.

SAEs have been successfully used to decompose LLM activations into interpretable features. This has been demonstrated in both toy models (Cunningham et al., 2023; Bricken et al., 2023) and state-of-the-art language models (Gao et al., 2024). SAEs can identify features ranging from concrete objects to abstract concepts (Templeton et al., 2024). While specific static concepts have been identified using SAEs, we explore the characteristics of in-context learning algorithms using this technique.

In the broader scope of RL and LLMs, several papers have investigated how the two methods can be integrated to improve LLMs. Le et al. (2022) coupled LLMs with deep RL in an Actor-Critic (Konda & Tsitsiklis, 1999) setup for program synthesis, where the deep RL model served as the critic for the code generated by the language models. Shinn et al. (2024) built a framework where an LLM received verbal RL feedback, allowing it to perform better in decision-making tasks, coding, and reasoning problems. Lastly, Brooks et al. (2022) developed a method for improving in-context policy iteration in LLMs for RL tasks. In contrast to these approaches where RL was used to improve LLM behavior, we focused on *understanding* the mechanisms through which an LLM can do RL. Given the demonstrated usefulness of RL in improving LLMs' abilities, it is essential to understand these mechanisms.

# 7 DISCUSSION

TD learning is a fundamental algorithm in artificial intelligence research (Sutton & Barto, 1987; 2018; Mnih et al., 2015). It offers a simple yet efficient solution to the problem of distilling temporally distant consequences of actions into an immediate value signal. TD learning is general: it can be used not only to learn future discounted rewards but also state occupancies (Dayan, 1993), uncertainties (Guez et al., 2012) and prediction errors (Burda et al., 2018; Saanum et al., 2024). It is therefore perhaps not surprising that neuroscience research has found traces of TD learning in striatal neural populations in several species. In this paper, we offer evidence that similar substrates of prediction and reward are implemented in the circuits of a large transformer network pretrained on internet-scale text corpora. This is surprising because the LLM was not trained using an RL objective but simply to perform next-token prediction. One hypothesis is that TD learning could facilitate next-token prediction by allowing the LLM to build more general models of the task. In the Graph Learning Task, the TD error can be used to build the SR which offers global geometric information about the graph structure. Representing features that span longer temporal horizons could be important for building rich representations of the world, which TD learning affords.

## 7.1 LIMITATIONS & FUTURE WORK

There are some important limitations and extensions of our work. First, our SAEs are task-specific and are not suitable for identifying RL-related variables for arbitrary tasks. Future work should attempt to build more generic SAEs and repeat our analyses with them. Perhaps, this can be accomplished through training SAEs on representations coming from a large and diverse corpus of RL tasks, or by fine-tuning pretrained SAEs (e.g., Lieberum et al. (2024)) on RL data. Similarly, to establish the generality of our results, future work could aim to replicate our findings using different LLMs.

Second, while we mapped out TD errors and $Q$-values across the residual blocks of Llama, a circuit-level understanding of how these representations come about remains unclear. Can induction heads (Olsson et al., 2022) give rise to these representations? Using methods such as path patching (Hanna et al., 2024; Wang et al., 2022), can in-context TD circuits be identified? These remain important questions for a complete understanding of how LLMs do in-context RL.

Lastly, while we can predict Llama's behavior using $Q$-learning agents and find TD-like representations in the model, their alignment is not perfect. Future work should identify which additional components are needed to fully account for the model's behavior. For example, adding a preference for repetition to the $Q$-learning model can be a natural extension to capture the tendency of pretrained LLMs to repeat themselves.

## 7.2 CONCLUSION

In this work, we have shown evidence that LLMs implement TD learning to solve RL problems in-context. Our work not only demonstrates that SAEs can disentangle features used during in-context learning but also that these discovered features can be used to manipulate LLM behavior and representations in a systematic fashion. Our finding paves the way for studying other types of in-context learning abilities using SAEs. Lastly, our work establishes a line of convergence between the learning mechanisms of LLMs and biological agents, who have been shown to implement TD computations in similar settings.

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

## A  Appendix

### A.1  $Q$-learning

In most RL settings we compute the TD error over a *batch* of observations. In offline or off-policy RL (Haarnoja et al., 2018), we calculate the TD error over a batch of state transitions experienced in the past, potentially collected using a different policy. To model TD learning in Llama, whose transformer architecture allows it to easily integrate information over multiple episodes in the past, we calculate the TD error as an average over the $k$ last state transitions, where $k$ is a hyperparameter. In the Two-Step Task, we use a $k = 4$, as this gave a good behavioral and representational fit. In the other tasks, we use the entire history of previous observations to compute the TD error, slightly smoothing out the error relative to $k = 4$. See Figure 7 for a qualitative comparison of the TD error with these settings. Other hyperparameters used to train the $Q$-learning model include the discount parameter $\gamma = 0.99$ across all tasks. The learning rate $\alpha$ was $0.1$ in the Two-Step Task and the Grid World, and $0.05$ in the Graph Learning Task. In the Two-Step task and in the Grid World, we passed the $Q$-value estimates through a softmax to obtain action probabilities, which were used to compute $-\log p(a)$.

In the Grid World task, we train the $Q$-learning agent using an $\epsilon$-greedy exploration strategy, where we linearly decay $\epsilon$ for the first $15$ episodes, leaving it at $0$ for the remaining episodes of the training run.

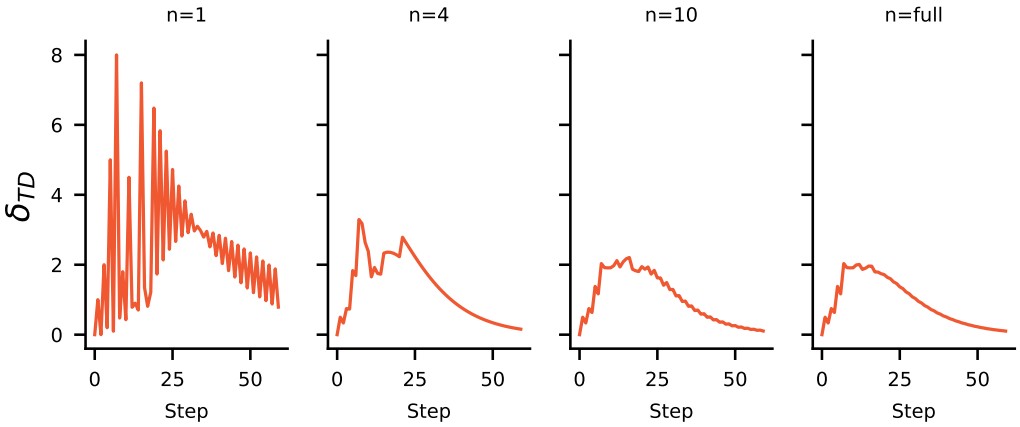

Figure 7: As the replay window $k$ grows, the TD signal is smoothed.

### A.2  SAE Training

For each task and each block of Llama, we train separate SAEs. For all SAEs, a batch size of $256$, a learning rate of $1e - 04$, and $\beta = 1e - 05$ were used. We used the Adam optimizer (Kingma & Ba, 2017) with the default parameters and shuffled the training data. The latent space for the Two-Step and the Grid World tasks was doubled in size. For the Graph Learning Task, we used an $8192$-dimensional latent space, equal to the input space size. We scaled our input data $\mathbf{H} \in \mathbb{R}^{n \times d}$ as follows before passing it through the SAEs: $\mathbf{H}' = \sqrt{d} \cdot \frac{\mathbf{H}}{\frac{1}{n} \sum_{i=1}^{n} |\mathbf{h}_i|_2}$. When we replaced Llama's activations with the reconstructed activity $\tilde{\mathbf{H}}'$ from the SAEs, we applied the inverse transformation. All the interventions were performed using the nnsight (Fiotto-Kaufman et al., 2024) library.

### A.2.1  SAE $L_0$ norms

Training SAEs on residual stream activations for the three experiments, we see that the original $8192$-dimensional representational space can be reduced to low-dimensional feature spaces with a few dozen active features. See Figure 8.

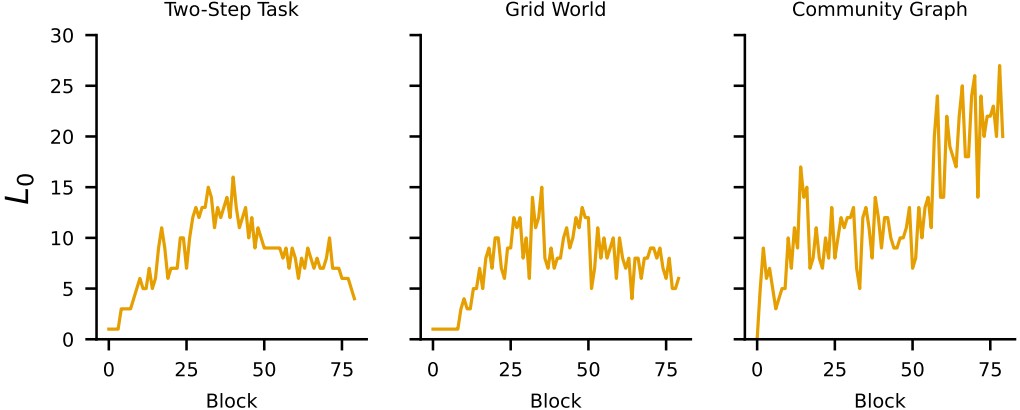

Figure 8: The $L_0$ norm of SAE representations (e.g. the number of features with non-zero variance) tends to peak in the middle blocks of Llama, except for in the Graph Learning Task, where it increases again towards the last blocks. This is presumably because Llama needs to predict the next token out of a larger set of token candidates in that task.

## A.3 THE TWO-STEP TASK

> **Prompt**
>
> You are an optimal reinforcement learning agent. You are playing a little game. You have two actions, Left and Right. Your goal is to get as many points as possible within an episode. Here we go.
>
> Episode 0: You pick right. You see an orange. You pick left. You get 6 points. Episode is over.
> [...]
> Episode 19: You pick left. You see an apple. You pick right. You get 9 points. Episode is over.

We used the prompt shown above for the Two-Step Task. Llama's internal representations were recorded at the pick tokens.

**Model comparison.** To test whether the negative log-likelihoods between the Myopic and the $Q$-learning were significant, we averaged negative-log likelihoods over each run and conducted a paired t-test. The $Q$-learning model was significantly better than the Myopic model ($t(99) = -3.40$, $p = 0.001$).

## A.4 REINFORCEMENT LEARNING IN A GRID WORLD

> **Prompt**
>
> What is the next object in the following sequence? You are living on a two-dimensional grid-world. You have 4 actions, left, right, up and down. Your goal is to get to the goal in as few steps as possible.
>
> Episode 0: You are at 0, 1. You go left. You get -1 points. You are at 0, 1. You go down.
>
> You get -1 points. You are at 0, 0. You go down [...]
>
> Episode 1: You are at 1, 1. You go right. You get -1 points. [...] You go right. You get +1 points. You finished with -3 points.

For the Grid World task, we prompt Llama as shown above. The `go` tokens are the recording points for internal representations, and the logits here were used to predict the action probabilities. This task was repeated 50 independent times, and we analyzed the behavior and the representations from the first 320 observations.

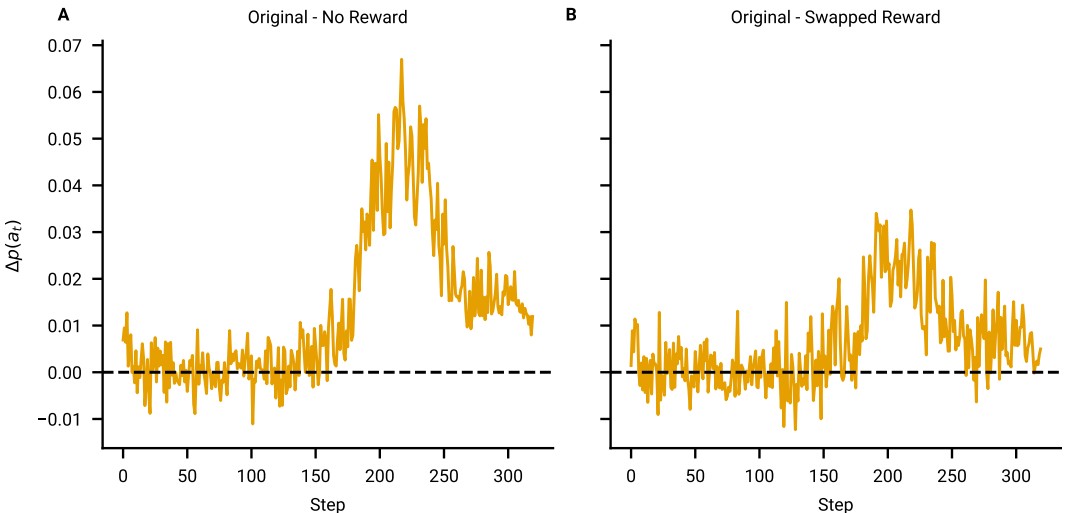

Figure 9: Additional control analyses for the Grid World. (*A*) The change in Llama's prediction accuracy of the $Q$-learning agent's actions from a control prompt with no rewards to the original prompt. (*B*) Same as *A*, except the control prompt uses sign-swapped rewards from the original prompt. Higher values in both plots indicate the effect of reinforcement during learning.

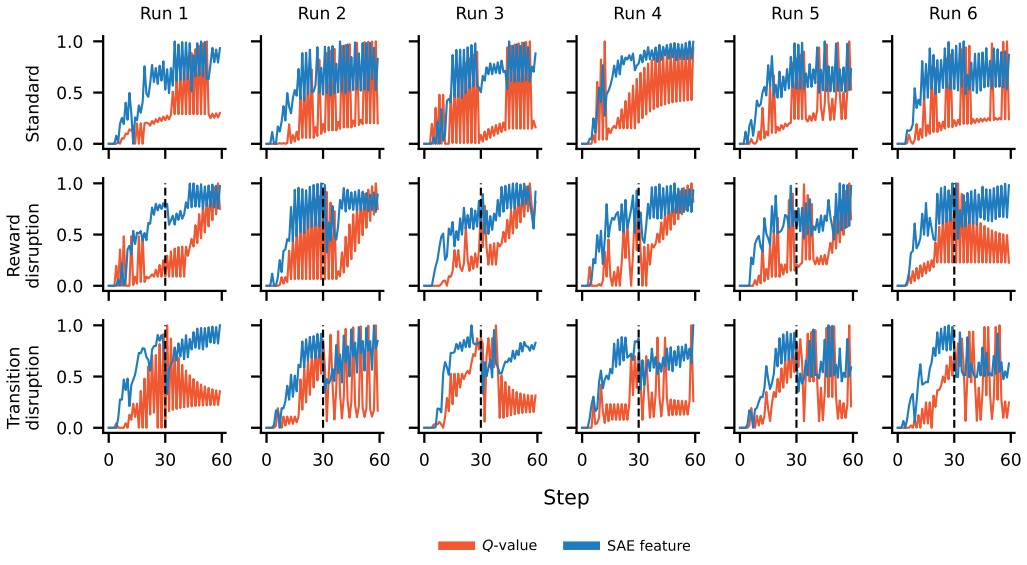

Figure 10: Examples of SAE features and $Q$-value for the Left action in the Two-Step task. There are dips both in the SAE feature and the $Q$-value at certain points that are late in the task. If Llama does TD-learning, this is to be expected when Llama ends up in a state where a particular action is suboptimal, just like the $Q$-value.

**Additional control analyses**. In addition to randomizing the rewards as described in the main text, we conducted two other control analyses. In the first one (Figure 9A), we removed the reward

information altogether from the prompt. During the exploration phase, this did not change how well Llama predicts actions, but we observed better predictions for the original prompt in later trials. In another control analysis, instead of randomizing the rewards, we swapped the $+$ and the $-$ signs (Figure 9B). This led to a smaller effect as removing the reward, but the shift was smaller in magnitude.

**Testing a simpler alternative to TD learning.** A reviewer raised the point that Llama's behavior could potentially be explained by a simpler mechanism than TD learning, for instance a model that keeps track of the rewards obtained at the end of each episode: Suppose in state $s$, it assigns an action a logit of $1$ if the action was performed in $s$ in the maximally rewarding episode. We then get the action probabilities by applying a softmax with a temperature parameter to the logits, where we fit the temperature parameter to minimize the negative log likelihood on Llama's behavior. We refer to this as the Look-up model. As can be seen in Fig. 11, the $Q$-learning model predicts Llama's choices better than the look up model.

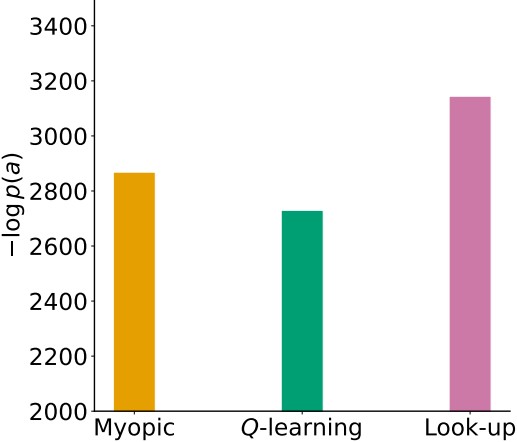

Figure 11: $Q$-learning fits Llama's behavior better than the Look-up model.

**A more difficult grid world.** We additionally tested Llama on a different variant of the grid world. In this setup, we used a 7 by 7 grid, increasing the state space from 25 to 49. Additionally, in each episode, we randomized the initial location of Llama, requiring strong generalization. The results are shown in Fig. 12. We found that Llama can learn to predict the $Q$-learning agent's actions in this task as well.

Using the larger grid world also lets us dissociate Llama's behavior from the Look-up model better. In the original grid world, we observed highly similar predictions between the two models. However, when we tested Llama on a larger grid world and initialized its starting position randomly in each episode, we observed that their predictions severely diverged, where Look-up's predictions stayed at around chance level. In the former environment, the sequences of actions become more deterministic over time, which is why Look-up is also a good model for that environment. However, changing the initial location of the model in each episode also changes the sequence of optimal actions. Llama can generalize to adapt to this, whereas Look-up cannot. The discrepancy in the new environment further suggests Llama is not implementing just Look-up but is indeed doing TD learning.

We also tested whether Llama's internal representations correlate more strongly with $Q$-values or Look-up values. As displayed in Fig. 13, in both environments, SAE features across blocks correlate more strongly with $Q$-values than with Look-up values. Notably, the differences in correlations are larger in the more difficult 7 by 7 Grid World (Fig. 13B), paralleling our behavioral findings. Taken together, both our behavioral and representational results show stronger evidence for Llama using TD learning than Look-up.

Lastly, we analyzed the models' predictions in states that don't occur at the start of the trajectories, but in the middle of other trajectories. Here the Look-up model predicts actions at random. Llama's action probabilities are not only considerably higher, but also match the action probabilities of the $Q$-learning agent with an $\epsilon$-greedy policy (Fig. 14).

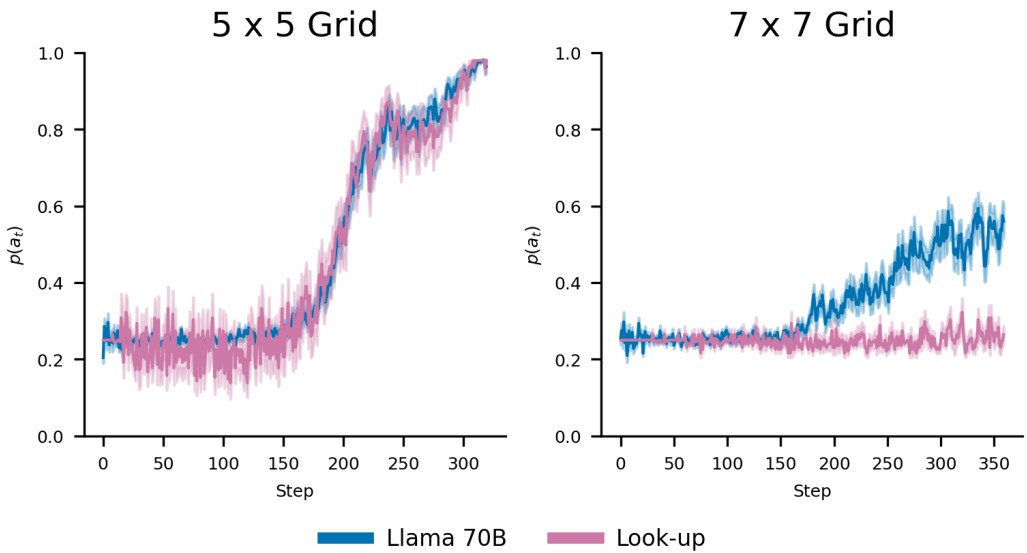

Figure 12: Llama learns to predict actions in larger grid-world environments. While Llama's predictions match those of the Look-up model in the small grid-world, this is not the case with larger grids which requires the agent to cache state-action values.

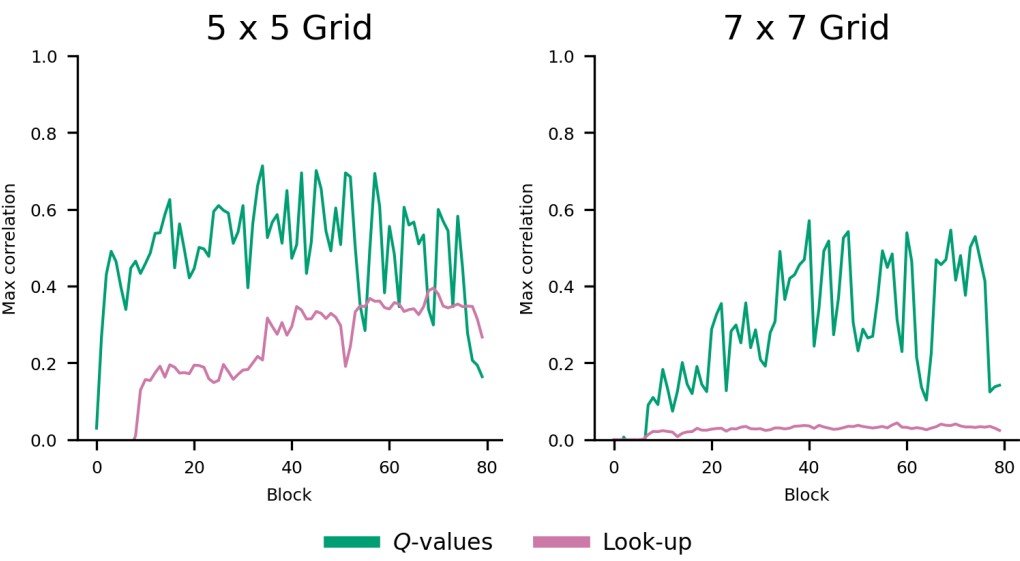

Figure 13: In both Grid World environments, SAEs show stronger correlations with $Q$-values than Look-up values.

## A.5   GRAPH LEARNING

**Prompt**

What is the next object in the following sequence?

cake , horse , gift , horse , woman , wall , sock , wall

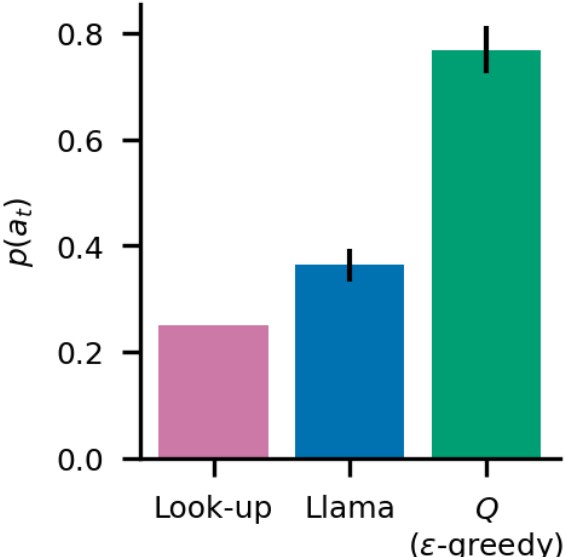

Figure 14: We analyze the models' predictions in states that have been visited before in the past, but from different starting states. Here the Look-up model predicts actions at random. Llama's action probabilities are not only considerably higher, but also match the action probabilities of the $Q$-learning agent with an $\epsilon$-greedy policy.

For the Graph Learning Task, we prompt Llama as shown above. Colors indicate how the text is tokenized. The representations corresponding to the ⟨,⟩ token are recorded, and we obtain Llama's predictions for the tokens that follow this. Recording representations always at the same token eliminates any differences in representations that may arise from using different tokens.

The latent community graph consisted of 3 communities, with 5 nodes in each community. Every node had 4 adjacent nodes. The node names are sampled from the category labels in the THINGS database (Hebart et al., 2019). We only sampled from labels represented as a single token in isolation, and the sampling was done with replacement across the 20 runs.

**Behavior.** On average, Llama had an accuracy of $24.3\%$ in predicting the next token, and the ceiling accuracy expected in the environment was $25\%$.

**Multidimensional scaling (MDS).** We used cosine distance to calculate the pairwise dissimilarity of representations both for Llama and the SR. This was done separately for different blocks of the transformer and the SR. For this analysis, we only considered the last encounter of both the Llama and the SR for each state. The dissimilarity matrices were used to project the representations onto a 2D space using metric MDS as implemented in `scikit-learn` (Pedregosa et al., 2011). What is plotted in Figure 5A is an example from one of the runs.

**Decoding bottleneck states.** We trained a linear support vector machine on Llama's state representations to predict whether the corresponding state is a bottleneck state. We did this in a leave-one-run-out fashion, where the data from each run served as the test data in one fold. This procedure also ensured that the state names could not be leveraged to aid classification, as they are randomly assigned across runs.

**Learning models.** We introduced some new learning models in this task that learn representations of the graph. First is the Successor Representation (SR)(Dayan, 1993), which is defined in the main text. We also consider an agent learning the transition matrix of the graph. This agent learns a representation $\mathbf{T}$, where $\mathbf{T}(s, s')$ is the estimated immediate transition probability from state $s$ to $s'$, which can be written as $\mathbf{T}(s, s') = \mathbb{E}[\mathbb{I}(s_{t+1} = s')|s_t = s]$. We learn this by counting the times the agent transitions from $s$ to $s'$, divided by the total number of transitions from $s$.

To accompany the transition probabilities, we also compute the *surprise* of a transition, as a measure of the *error* of the learned model.

$$\text{surprise}(s_t, s') = \begin{cases} -\log \mathbf{T}(s_t, s'), & \text{if } s' = s_{t+1} \\ -\log(1 - \mathbf{T}(s_t, s')), & \text{otherwise} \end{cases} \tag{6}$$

This yields a vector-valued signal surprise$(s, :)$ for a given state $s$.

**SAE analyses.** The SAEs were trained for 20 epochs with an $8192$ dimensional latent space, which was identical to the input space size. After training the SAEs for each transformer block, we correlated the activity of the SAE latents with non-zero variance against estimates of our learning models. For each model, we stack representations row-wise and obtain a step-by-state matrix. We then correlate every column of this matrix with every SAE unit. Then, for each model and each state, we take the maximum correlation and report the average of this across runs.

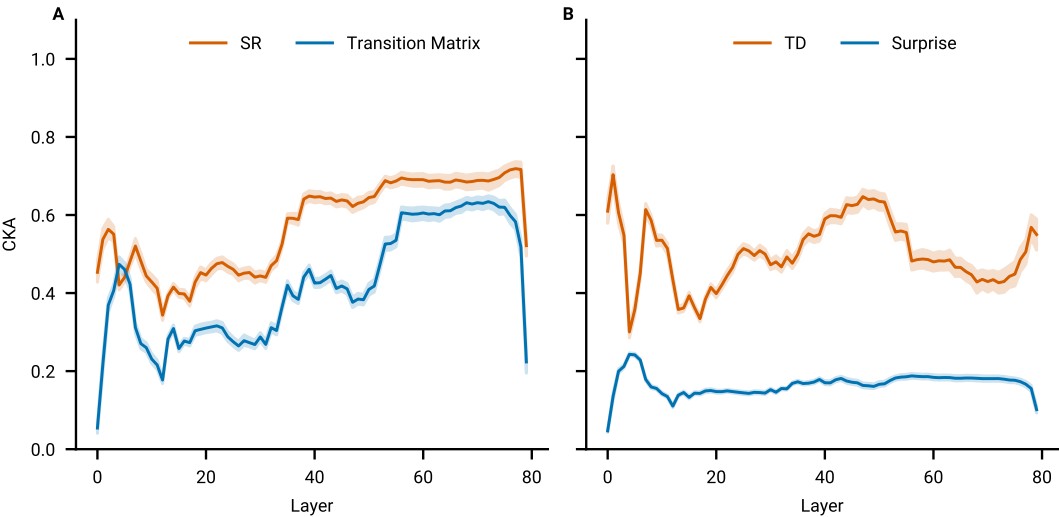

Figure 15: Replication of Graph Learning SAE results using CKA. SR (TD) is more similar to Llama than the learned transition matrix (surprise).

**Replication using representational similarity analysis (RSA).** Comparing vector-valued model representations across two different spaces has traditionally been done using RSA both in neuroscience (Kriegeskorte et al., 2008) and machine learning (Yousefi et al., 2024). Here, we replicate our SAE findings using RSA. Specifically, we used a variation of RSA called Centered Kernel Alignment (Kornblith et al., 2019) (CKA) that is commonly used in machine learning, where similarity is calculated between representations $\mathbf{X}$ and $\mathbf{Y}$ as follows:

$$\text{CKA}(\mathbf{X}, \mathbf{Y}) = \frac{||\mathbf{Y}^T \mathbf{X}||_F^2}{||\mathbf{X}^T \mathbf{X}||_F ||\mathbf{Y}^T \mathbf{Y}||_F} \tag{7}$$

where $|| \cdot ||_F$ denotes the Frobenius norm. CKA is bounded between $0$ and $1$, and higher CKA indicates stronger similarity. We stacked Llama's and the learning models's representations during the task row-wise and calculated the CKA between them for each run. As shown below, we arrive at the same conclusion using this method: Llama represents the environment like the SR and learns this representation through TD learning.

## B  APPENDIX: REPLICATIONS WITH OTHER LLMS

To test whether our results generalize to other LLMs outside the Llama family, we repeated the majority of our analyses for all three tasks (Two-Step, Grid World, and Graph Learning) on two new

models, Gemma-2-27B (Team et al., 2024) and Qwen-2.5-72B (Qwen Team, 2024). Remarkably, almost all analyses yielded qualitatively matching results, showing a high degree of similarity in how these models solve RL tasks in-context. We discuss each individual point raised in the review in more detail below.

We observed that neither model can do the Two-Step task as well as Llama. Consequently, the RL models predict behavior not as accurately for these models. However, the SAE correlations remain similar to the original findings. The results are visualized in Fig. 16 and Fig. 17 respectively. For the Grid World, the replications closely match the original findings. We visualize them in Fig. 18 and 19. Both models can learn to predict the actions in the sequence, and we observe stronger correlations in the SAE features with TD driven value and error signals compared to alternatives. Similarly, in the Graph Learning task, both models have strong SR and the community structures emerge over the transformer blocks. These findings are visualized in Fig. 20 and Fig. 21.

**Gemma-2-27B**

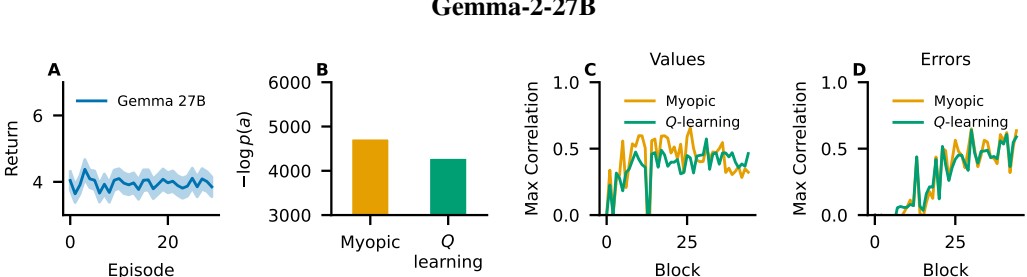

Figure 16: For Gemma-2-27B, there are SAE features that correlate highly with TD errors and values. The model can not do the task as well as Llama. Consequently, the RL models predict behavior not as well for these models

**Qwen-2.5-72B**

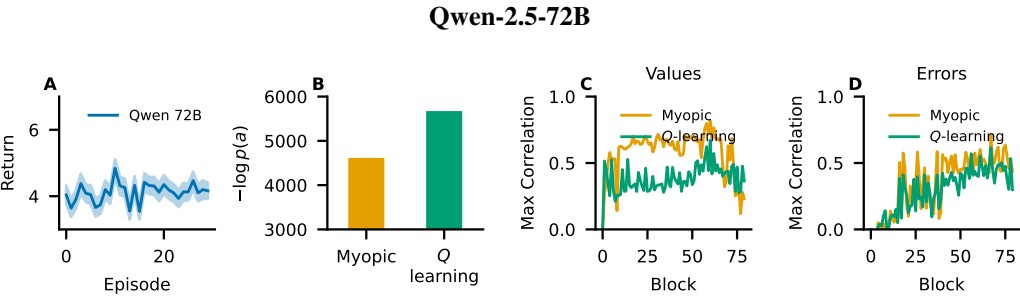

Figure 17: For Qwen-2.5-72B, there are SAE features that correlate highly with TD errors and values. The model can not do the task as well as Llama. Consequently, the RL models predict behavior not as well for these models

**Gemma-2-27B**

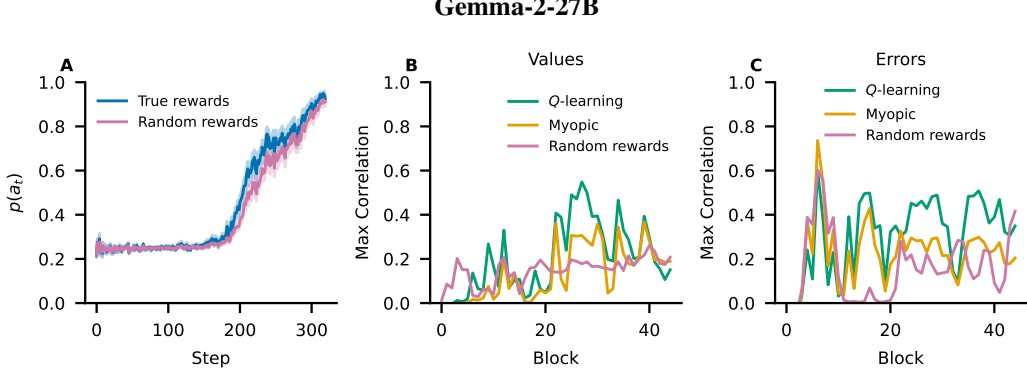

Figure 18: Behavioral and SAE correlation analyses reported in Fig. 4 replicate with Gemma-2-27B.

**Qwen-2.5-72B**

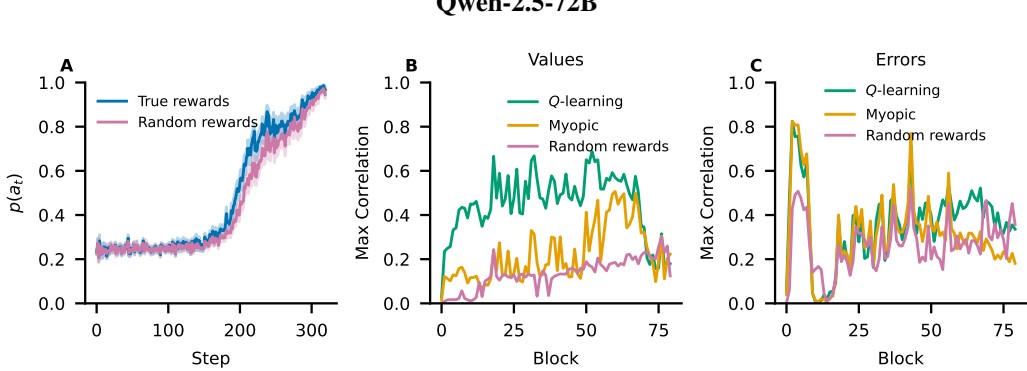

Figure 19: Behavioral and SAE correlation analyses reported in Fig. 4 replicate with Qwen-2.5-72B.

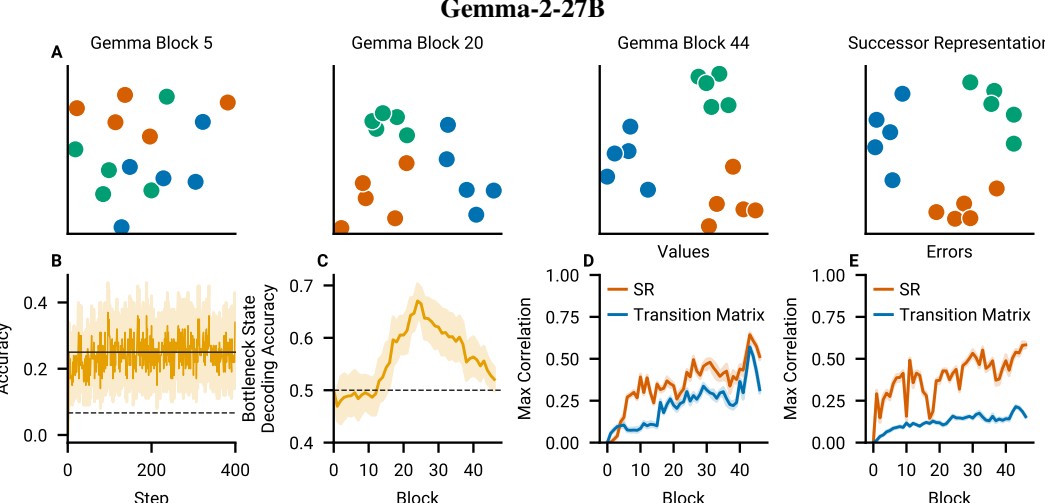

Figure 20: Behavioral, decoding, and SAE correlation analyses reported in Fig. 5 replicate with Gemma-2-27B.

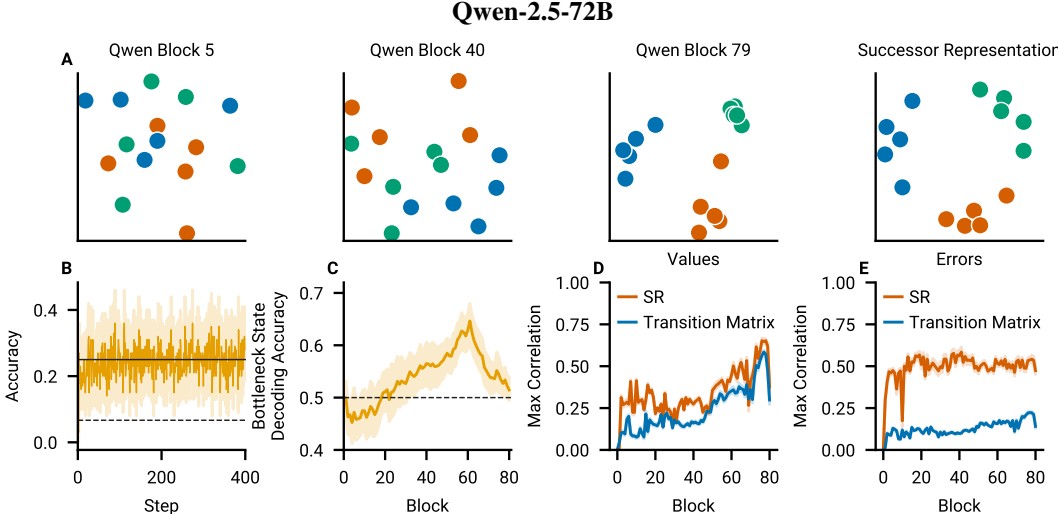

Figure 21: Behavioral, decoding and SAE correlation analyses reported in Fig. 5 replicate with Qwen-2.5-72B.

