# OpenReview forum: "Sparse Autoencoders Reveal Temporal Difference Learning in Large Language Models"
_ICLR.cc/2025/Conference — ICLR 2025 Poster_

### Official Review · Reviewer_VvRq · 2024-11-01

**Soundness:** 2
**Presentation:** 4
**Contribution:** 3
**Rating:** 6
**Confidence:** 4

**Summary:**

The paper looks for evidence of whether Llama3-70B model can simulate the TD-learning algorithm for solving RL tasks. The authors evaluate this using simple toy RL environments. They train different SAEs for the different tasks and find features that correlate highly with TD-errors and Q-values. They confirm that these features are causal in predicting actions by performing interventions on such features. Based on these evidence, the authors conclude that the LLM is simulating the TD-learning algorithm.

**Strengths:**

1. The study evaluates their hypothesis through a series of tasks to substantiate their empirical claims.
2. Intervention experiment with the features to confirm their causal roles.
3. The writing is clear and easy to understand. However, some details are missing. See in questions.

**Weaknesses:**

My main objection with the paper is that there is a simpler alternative hypothesis that could equally explain all of the results. Given the simplicity of the task, the LLM could be implementing the following algorithm to solve the tasks:


Step 1: Keep a track of the maximum points for each episode in the context.

Step 2: Predict the actions from the episode that has the maximum points.


This algorithm is simple to implement for the LLM given previous works on LLMs implementing greater than circuit [1] and induction heads [2]. Also, for the Two-Step Task, the first 7 episodes are provided by using a random policy, which should cover all the 4 different trajectories possible in the task.

The features that the authors find using SAEs could be features that are tracking the maximum points across episodes. These features will have high correlation with Q-values, and are also causal so interventions on them should show similar results as shown in the paper.

I recommend that the authors conduct experiments designed to refute this hypothesis. See questions for some suggestions on experiments that can be performed.

References:

[1] How does GPT-2 compute greater-than?: Interpreting mathematical abilities in a pre-trained language model. https://arxiv.org/abs/2305.00586

[2] In-context Learning and Induction Heads. https://arxiv.org/abs/2209.11895

**Questions:**

1. In the plots on max correlation with values/errors (eg fig 2c, 2d, 3, 4b, 4c, etc.), is the correlation computed with the value/error of the action predicted by the LLM at the given state? If yes, then it would be valuable to check whether there are features that correlate with value/error of non-optimal actions. This could help in distinguishing whether the LLM is actually implementing TD-learning or the max-point episode algorithm provided above.
2. Can you provide how the NLL score computed? I couldn't find it in the appendix either. Particularly, are you computing the log probabilities of Q-learning agent by doing a softmax using the Q-values over actions?
3. Are you using any discount rate for the Grid World Task? If yes, please provide it.

---

> ### Author Response · Authors · 2024-11-21
>
> >My main objection with the paper is that there is a simpler alternative hypothesis that could equally explain all of the results. Given the simplicity of the task, the LLM could be implementing the following algorithm to solve the tasks:
>
> > Step 1: Keep a track of the maximum points for each episode in the context.
>
> > Step 2: Predict the actions from the episode that has the maximum points.
>
>
> We would like to thank the reviewer for engaging with our paper and for the constructive feedback. The reviewer raised an important point about alternative explanations for the results of the paper. We agree that the proposed mechanism could in principle account for some of our results and therefore conducted new experiments and analyses to test this alternative account. We find that TD learning predicts our results much more accurately than the alternative account proposed by the reviewer (which we called Look-up). Details as well as answers to the other questions are presented below.
>
>
> In the Two-Step Task, we implemented Look-up with a softmax decision rule (in state $s$, it assigns an action a logit of 1 if the action was performed in s in the maximally rewarding episode), where we fitted a temperature parameter to the softmax to minimize the negative log-likelihood of Llama’s actions. We found that the Look-up model does not predict Llama’s behavior as well as a $Q$-learning agent (Page 19, Figure 11).
>
> We also implemented the Look-up model for the Grid World task. Fitting it to the actions produced by the $Q$-learning agent, we see that its ability to predict actions matches Llama’s ability to predict actions (page 20, Figure 12 left). To further investigate this we tested Llama on a larger Grid World with 49 states (page 20, Figure 12 right), almost doubling the state space from the small Grid World. We also randomly initialized the starting position every episode, meaning that Llama had to generalize to predict the $Q$-learning agent’s actions. Here Llama’s action predictions diverged strongly from the Look-up model, indicating that it relied on a different mechanism than pure look-up to predict actions, which is likely TD learning.
>
> Lastly, we showed that the Grid World SAEs showed much stronger correlations with $Q$ values compared to the Look-up’s estimates (Page 20, Fig. 13 left). Notably, these differences were further amplified in the new Grid World (Page 20, Fig. 13 right), paralleling our behavioral findings. Taken together, both our behavioral and representational results show stronger evidence for Llama using TD learning than Look-up.
>
>
> > (...) is the correlation computed with the value/error of the action predicted by the LLM at the given state?
>
> We compute the correlations between the Q/TD of each action and the SAE features separately. What we plot is the average of the correlations across the different actions.
>
> > if yes, then it would be valuable to check whether there are features that correlate with value/error of non-optimal actions.
>
> Please see Figure 10 in the Appendix, where we plot an example $Q$-value latent and the $Q$ value estimated by the RL model. We observe dips both in the SAE feature and the $Q$-value at certain points that are late in the task. If Llama does TD-learning, this is to be expected when Llama ends up in a state where a particular action is suboptimal, just like the $Q$-value. This cannot be explained by the Look-up model.
>
>
>
> > Can you provide how the NLL score computed? I couldn't find it in the appendix either. Particularly, are you computing the log probabilities of Q-learning agent by doing a softmax using the Q-values over actions?
>
> Thanks for pointing out this missing detail. Indeed, we do a softmax over the $Q$-values to compute the log probabilities. This is now added to the appendix on page 16.
>
> > Are you using any discount rate for the Grid World Task? If yes, please provide it.
>
> Yes, we use $\gamma = .99$ for all tasks. We have made this clearer in the appendix Line 822 now.

---

> > ### Comment · Reviewer_BhvT · 2024-11-25
> >
> > I just wanted to say that I really appreciated Reviewer VvRq's comment, as well as the authors' response. These new experiments definitely strengthen the paper in my view.

---

> > ### Comment · Reviewer_VvRq · 2024-11-25
> >
> > Thank you for the response and conducting the additional analysis in the paper.
> >
> > The Look-Up model you described makes sense. The newer results on the 5x5 and 7x7 grid-world task are also interesting. It is obvious that the Look-Up policy (as described) will perform badly when you initialize the agent from different positions. I would guess that Llama is not implementing the basic Look-Up that you tested but is doing the look-up by also conditioning on the initial state. That is, it is picking the action from the episode that got the highest return **and** has the same initial state. I would be very interested to see the authors report the results with this State-Look-Up policy instead.
> >
> > Experiment to test: The State-Look-Up policy should perform poorly when tested on an initial grid-position that doesn't occur at the start of the in-context trajectories but does occur in the middle of some trajectories. In this case, the Q-learning policy should still perform optimally. This experiment will reject my hypothesis that Llama is implementing the State-Look-Up policy if it performs optimally.
> >
> > I am increasing my rating from 3 -> 5 because the authors did refute the basic Look-Up policy I originally proposed. I am still a bit sceptical of the claim that Llama is implementing Q-learning (mostly because it is an extraordinary claim that requires extraordinary evidence). The results from the above experiment will update me more towards Llama implementing Q-learning. Hence, I am still open to increase my rating if the authors conduct the proposed experiment or some variant that refutes the State-Look-Up policy.

---

> ### Author Response · Authors · 2024-11-25
>
> Thank you for your response and the new suggestion. We have modified the Look-up model so that it conditions on the initial state when predicting actions, just like the reviewer proposed. We see that this model’s action predictions do not match Llama’s action predictions in the 7x7 Grid World either (see the updated Figure 12, page 20). Since the state-space is combinatorially large, the Look-up mechanism cannot predict the $Q$-learning agent’s actions nearly as well as Llama, suggesting that Llama maintains representations of cached $Q$-values. Furthermore, we still see that $Q$-values correlate much more strongly with SAE features than the Look-up action-values (see updated Figure 13, page 20).
>
> We also followed the reviewer’s suggestion to analyze the models’ predictions in states that don't occur at the start of the trajectories, but in the middle of other trajectories. Here the Look-up model predicts actions at random. Llama’s action probabilities are not only considerably higher, but also more closely match the action probabilities of the $Q$-learning agent with an $\epsilon$ -greedy policy (see Figure 14, page 21).
>
> We hope this addresses the reviewer’s concern.

---

> > ### Author Response · Authors · 2024-11-29
> > **Gentle reminder**
> >
> > We would like to gently remind the reviewer to review our last analyses. We hope these analyses and our previous response have answered the reviewer's questions. We are happy to answer any outstanding questions the reviewer may have. If all concerns have been addressed, we would appreciate if the reviewer would consider raising their score.

---

> > ### Comment · Reviewer_VvRq · 2024-11-29
> >
> > Thank you for conducting the suggested experiments! The results definitely point in favor of Llama doing something similar to TD-learning.
> >
> > I have updated my score to 6. I am still not fully convinced with the claim that LLMs implement TD learning to solve RL problems. The SAE feature based correlation & causal analysis is not fine-grained enough to justify this claim. There could be other simpler algorithms that have internal variable that correlate with TD-learning. For example, the authors show that the Look-Up policy is an equally good explanation for 5x5 grid but fails at 7x7 grid. It could be possible that TD-learning is a good explanation for the toy tasks considered but there is another algorithm that might equally (or better) explain the current results. I think a fine-grained circuit-level analysis would be needed to justify the claim of LLMs implementing TD-learning, although I understand that circuit-level analysis is not in the scope of this paper. I am only pointing out that further research will need to be done in order to make the strong claim of the paper.

---

### Official Review · Reviewer_fLyU · 2024-11-02

**Soundness:** 2
**Presentation:** 2
**Contribution:** 2
**Rating:** 5
**Confidence:** 1

**Summary:**

This paper is way out of my expertise and hence I cannot provide a meaningful review.

**Strengths:**

.

**Weaknesses:**

.

**Questions:**

.

---

### Official Review · Reviewer_H3vx · 2024-11-02

**Soundness:** 3
**Presentation:** 3
**Contribution:** 3
**Rating:** 6
**Confidence:** 3

**Summary:**

The paper presents a mechanistic analysis of internal activations of the Llama 3 70b model during three different in-context reinforcement learning tasks. In particular, the authors use Sparse Auto-Encoders (SAEs) to generate latent space representations of the residual streams and show how they correlate with TD Errors and Q-values of Q-learning agents trained to solve the same tasks. Furthermore, they show that relationship between the latent representations and the TD Errors is causal by intervening on the latent representations, which causes a degrading in the performance of the model.

**Strengths:**

I think the paper is well written and the setting and the experimental details are generally well explained. The contributions are also clearly stated. Furthermore, as far as I can tell, the presented experimental methodology is also sound. Although it is a known fact that Transformers can potentially perform In-Context RL, especially if trained for it, it is the first time, to the best of my knowledge, that a mechanistic analysis is conducted on a model which was pre-trained on next token prediction. In addition, even if the methods used (e.g. SAEs) are already well established in the mechanistic interpretability literature, it is insightful to see how they can be successfully used also to better understand how LLMs solve In-Context RL. Hence, even if the problem of In-Context RL is well studied in the literature and the interpretability methods used are also well established, overall I think the presented results shed more light on the inner workings of how LLMs can solve RL tasks in-context, which can be significant and insightful for the community.

**Weaknesses:**

- The main weakness of the paper is that being an experimental work, I find the number of experiments conducted to be a bit limited. I think that more experiments should be conducted to further support the contributions of the paper (I saw that the authors mention this in future works/limitations, but I think the current paper would benefit from more ablation to make the evidence stronger). In particular, I suggest that the authors (as they also mention) should try to repeat the experiments they present with different models (at least one more) to prove that their results hold in general for "big enough" models. This would be really insightful since it would tell us that different models, even if trained differently, learn similar representations or make use of similar strategies to solve tasks. Furthermore, I think it would be insightful to conduct experiments on larger environments to better understand both to what extent these models are capable of performing In-Context RL and to analyze if, even at larger scale, these models still make use of TD Erros and Q-Values to solve the task
- One minor concern regards the extent of the novelty of the work: as I mentioned above, although I agree with the authors that it is the first time (to the best of my knowledge) that it was shown that models trained on next-token prediction perform In-Context RL exploiting TD Errors, there are already quite some works exploring TD Learning in Transformers (both at a theoretical and experimental level). Furthermore, the methodology used for the mechanistic analysis is also already well established in the mechanistic interpretability literature.

**Questions:**

Some small additional comments and questions I had:
- In the definition of the Q function in Section 2 (Methods, at page 2), shouldn't there be a conditioning on the initial state and action inside the expectation? Also, shouldn't the sum start from $t=0$ instead of $t=1$?
- In Section 3, you claim that Llama 3 most likely implements "classic" Q-Learning rather than myopic Q-learning based on the negative log-likelihood. However, in Figure 2, looking at the correlations, it seems that the myopic Q-learning has in general comparable if not higher correlations to the latent representations. Couldn't this suggest that the model is implementing the myopic algorithm instead? Furthermore, is the difference in negative log-likelihood statistically significant?
- In Figure 5, what do the horizontal lines in subplots B & C represent?

---

> ### Author Response · Authors · 2024-11-21
> **Part 1**
>
> We thank the reviewer for their thoughtful review. We are glad the reviewer found our paper well-written and our methodology sound. The reviewer raised some important points, particularly, if our results generalize to other LLMs outside the Llama family. To address this, we repeated the majority of our analyses for all three tasks (two-step task, grid world, and graph learning) on two new models, Gemma-2-27b and Qwen2.5-72B. Notably, almost all analyses yielded qualitatively matching results, showing a high degree of similarity in how these models solve RL tasks in-context. We discuss each individual point raised in the review in more detail below.
>
> >in particular, I suggest that the authors (as they also mention) should try to repeat the experiments they present with different models (at least one more)
>
> Thanks for this suggestion. We have repeated all three experiments using two new models (Gemma-2-27b and Qwen2.5-72B). We place these findings in the Appendix, and here is a summary of the results:
>
> - Two-Step Task (Page 23 Figures 15 and 16): For both Qwen and Gemma, we find SAE latents that correlate highly with TD errors and values. However, neither model can do the task as well as Llama. Consequently, the RL models predict behavior not as well for these models.
> - Grid World (Page 24, Figures 17 and 18): The results are qualitatively the same. $Q$-learning values and error signals are more strongly correlated with SAE features than the value/error signals of the myopic model. We found both models can learn to predict actions from rewards.
> - Graph Learning (Page 25, Figures 19 and 20): The results are qualitatively the same. Strong SR and TD correlations are found in both of these models, and the community structures emerge over the transformer blocks.
>
> > Furthermore, I think it would be insightful to conduct experiments on larger environments to better understand both to what extent these models are capable of performing In-Context RL
>
> We tested Llama on a new grid world with two important differences from the grid world we initially tested:
> The new environment had $49$ states, almost double the size of the grid world we initially tested, which had $25$ states.
> In each episode, we randomized the initial location of Llama, requiring strong generalization.
>
> The results are shown on page 20, Figure 12. We found that Llama can learn to predict the Q learning agent’s actions here as well, though performance is a bit weaker in this more challenging task. Importantly, since the starting position is randomly initialized in each episode, Llama cannot solely rely on looking up what the $Q$-learning agent did in the past to predict what it will do in the future. Furthermore, we observed strong correlations between TD/$Q$-values and SAE features, as shown on page 20 Fig. 13. These findings provide further evidence that Llama uses TD errors and $Q$-values in larger environments as well.
>
> > One minor concern regards the extent of the novelty of the work (...) although I agree with the authors that it is the first time (to the best of my knowledge) that it was shown that models trained on next-token prediction perform In-Context RL exploiting TD Errors, there are already quite some works exploring TD Learning in Transformers
>
> Past works have indeed shown that Transformers can implement RL algorithms when trained either to solve RL tasks directly or to predict action sequences produced by agents trained with an explicit RL objective (e.g. behavioral cloning) [1, 2]. However, training LLMs differs significantly from these training setups: LLMs are multi-billion parameter models trained on internet-scale text data using next-token prediction as its objective, whereas past work has investigated small-scale Transformers trained directly on state-action-reward triplets. We find LLMs doing TD learning surprising, given the major differences in their training compared to the other transformer models discussed. Moreover, past works have not investigated the mechanistic basis of the learning algorithms these Transformers implement after training. Our paper fills this gap by examining the mechanisms of in-context RL in LLMs.  We added these points and references to the Related Work section.
>
>
> > Furthermore, the methodology used for the mechanistic analysis is also already well established in the mechanistic interpretability literature.
>
> Indeed, SAEs and targeted interventions have seen wide applications in recent years in terms of uncovering the inner workings of LLMs, which is why we chose these methods. An important novelty of our work is that we use SAEs to identify learning algorithms implemented in-context, in contrast to identifying static concepts. This marks a novel use case of these models and demonstrates how they can be used to understand in-context learning better.

---

> > ### Author Response · Authors · 2024-11-21
> > **Part 2**
> >
> > > In the definition of the Q function in Section 2 (Methods, at page 2), shouldn't there be a conditioning on the initial state and action inside the expectation? Also, shouldn't the sum start from t=0  instead of  t=1?
> >
> > The equation defines the value of a state-action pair in a Markovian setting and is therefore independent of the initial state. We have also changed the sum to go from t=0 on line 103.
> >
> > > In Section 3, you claim that Llama 3 most likely implements "classic" Q-Learning rather than myopic Q-learning based on the negative log-likelihood. However, in Figure 2, looking at the correlations, it seems that the myopic Q-learning has in general comparable if not higher correlations to the latent representations. Couldn't this suggest that the model is implementing the myopic algorithm instead?
> >
> > Thanks for raising this point. The myopic values simply track how immediately rewarding a certain state-action pair is, and are therefore not mutually exclusive with tracking $Q$-values. Since rewards are provided in the prompt, it makes sense that Llama develops representations that are predictive of the reward, as tracking myopic values is a prerequisite for tracking $Q$ values. However, tracking them internally does not necessarily mean they directly control behavior. Indeed, that is what behavioral model fitting shows us, that the classic $Q$-values drive Llama’s choices.
> >
> > >is the difference in negative log-likelihood statistically significant?
> >
> > Yes. We conducted a t-test between the negative log-likelihoods averaged over each run and found that the $Q$-learning model fits the data significantly better than the myopic model ($t(99) = 3.40$, $p = .001$). We have added this comparison to the appendix as well.
> >
> > > In Figure 5, what do the horizontal lines in subplots B & C represent?
> >
> > Dashed horizontal lines indicate chance level performance and the solid horizontal line represents the ceiling. Thanks for catching this. We updated the figure caption with the explanations.
> >
> > [1] Laskin, Michael, et al. "In-context reinforcement learning with algorithm distillation." arXiv preprint arXiv:2210.14215 (2022).
> >
> > [2] Wang, Jiuqi, et al. "Transformers Learn Temporal Difference Methods for In-Context Reinforcement Learning." arXiv preprint arXiv:2405.13861 (2024).

---

> > > ### Comment · Reviewer_H3vx · 2024-11-25
> > >
> > > I thank the Authors for conducting the additional experiments and for clarifying my doubts, I am satisfied by their answers and hence I am still willing to recommend the paper for acceptance.

---

### Official Review · Reviewer_BhvT · 2024-11-03

**Soundness:** 4
**Presentation:** 3
**Contribution:** 4
**Rating:** 8
**Confidence:** 4

**Summary:**

The paper investigates whether Llama 3 70B has internal representations that support temporal difference learning. First, it demonstrates that Llama can solve RL tasks significantly better than chance. Next, it trains a sparse autoencoder (SAE) and finds features correlated with TD error. Finally, it causally intervenes on these features to show that in-context RL performance degrades without those specific TD features.

**Strengths:**

This is an excellent paper. It asks a very interesting question and provides compelling evidence for the conclusion that Llama represents TD error and uses it to solve RL problems in-context. The section on successor representations was a welcome surprise in section 5, and offered more evidence for TD learning, even absent any rewards. The paper was also quite easy to follow and laid out the argument in a very natural way. I don't have any major complaints.

**Weaknesses:**

Only minor weaknesses.

1. In the background section on RL, TD is presented for a fixed policy, and then the paper switches to Q-learning, assuming the policy chooses \argmax_a Q(s,a). But this will change the policy as the Q function is updated, so it's not technically the same setting.
2. It was a bit unclear what "control lesion" referred to in Fig. 2F. And more generally, I was not familiar with the "lesion" terminology, so a brief definition would be welcome. I assume it's a form of activation patching?
3. I would have liked slightly more explanation regarding "clamping" the activations. I assume this means setting them to a specific value, but how is that different from deactivating them (i.e. clamping them to zero)? Is the purpose of clamping the activations to show degraded, unchanged, or improved performance?
4. Line 458, mangled sentence "our study is, we have explored".

**Questions:**

Could you please provide clarification re: weaknesses 2 & 3?

---

> ### Author Response · Authors · 2024-11-21
>
> We would like to thank the reviewer for engaging with our work and for their encouraging feedback. We are happy that the reviewer thought our paper was excellent and easy to follow. The reviewer raised some minor points which we have addressed below.
>
>
> > In the background section on RL, TD is presented for a fixed policy, and then the paper switches to Q-learning, assuming the policy chooses \argmax_a Q(s,a). But this will change the policy as the Q function is updated, so it's not technically the same setting.
>
> Thanks for raising this point. We have added the following clarification to our paper on line 119:
>
>
> **“For subsequent analyses, we rely on $Q$-learning, which is a variant of TD learning that learns a value function for the optimal policy.”**
>
>
> >It was a bit unclear what "control lesion" referred to in Fig. 2F. And more generally, I was not familiar with the "lesion" terminology, so a brief definition would be welcome. I assume it's a form of activation patching?
>
> In a control lesion, we set the latent unit that has the lowest correlation with Q/TD from a given layer to $0$. We have added the following clarification on line 330:
>
> **“We also conducted control lesions, where we set the activity of a latent unit from the same block with the lowest TD correlation to $0$.”**
>
> Lesioning means that we set the activation of a particular unit to 0. We have added the following clarification:
>
> **“Lesioning refers to setting the activations of specific units to $0$. This is also commonly referred to as zero ablation in the literature [1].”**
>
> >I would have liked slightly more explanation regarding "clamping" the activations. I assume this means setting them to a specific value, but how is that different from deactivating them (i.e. clamping them to zero)? Is the purpose of clamping the activations to show degraded, unchanged, or improved performance?
>
> Thank you for raising this point. By clamping, we refer to multiplying the activity with a scalar. We replaced this term with scaling in the main text to improve clarity. The purpose of the negative scaling analyses was to show degraded Q/TD representations in subsequent blocks, as shown in Fig. 2G and Fig. 2H
>
> >Line 458, mangled sentence "our study is, we have explored".
>
> Thanks, we corrected the sentence:
>
> **“While specific static concepts have been identified using SAEs, we have explored the characteristics of in-context learning algorithms using this technique.”**
>
> [1] Heimersheim, S., & Nanda, N. (2024). How to use and interpret activation patching. arXiv. https://doi.org/10.48550/arxiv.2404.15255

---

> > ### Comment · Reviewer_BhvT · 2024-11-26
> >
> > Great. Thanks for the response! I like these changes. I'm familiar with activation patching and zero-ablation; I just hadn't heard of "lesioning" before.

---

### Author Response · Authors · 2024-11-21
**General Response**

**edit: after the rebuttal, all reviewers recommend acceptance with an average score of 6.67 (again discounting one empty review). We thank all reviewers for their engagement during the review process.**

We would like to thank all reviewers for their constructive interactions. The overall assessment was positive with an average rating of 5.67 pre-rebuttal (discounting one empty review).

There was one reviewer (fLyU) who gave a score of 5 without providing a review, stating that they were unable to assess the paper due to lacking expertise. We expect that this review will not be considered when making a final decision. Besides that:

- Reviewer BhvT mentioned that “this is an excellent paper.”
- Reviewer H3vx said that our results “can be significant and insightful for the community.”
- Reviewer VvRq highlighted that the “writing is clear and easy to understand.”

In response to the reviewers’ feedback, we have made the following major modifications to our manuscript:

- We have replicated most of our results for two additional models (Gemma-2-27b and Qwen2.5-72B) as suggested by reviewer H3vx. We find evidence for TD representations in both these models.
- We have extended our results to a larger grid world environment with random starting states (as requested by reviewer H3vx).
- We have verified that the model’s behavior and internal representations do not match a simpler memorization and look-up strategy as suggested by reviewer VvRq.

We describe these and other smaller changes in detail in our responses to the individual reviews below. We again want to thank the reviewers for their valuable input, we believe it has substantially improved our paper.

---

### Meta-Review · Area_Chair_a9Tk · 2024-12-23

**Metareview:**

The paper investigates whether Llama 3 70B has internal representations that support temporal difference learning. First, it demonstrates that Llama can solve RL tasks significantly better than chance. Next, it trains a sparse autoencoder (SAE) and finds features correlated with TD error. Finally, it causally intervenes on these features to show that in-context RL performance degrades without those specific TD features.

I personally think this sort of analysis is interesting and very relevant to the community. It is also interesting for future work to explore the reasons for emergence of this sort of TD learning since that will be very nice too!

The reviewers all liked the paper and voted for accepting it, hence the paper is being accepted.

**Additional Comments On Reviewer Discussion:**

See above.

---

### Decision · Program_Chairs · 2025-01-22

Accept (Poster)